# Allelic Variations in Phenology Genes of Eastern U.S. Soft Winter and Korean Winter Wheat and Their Associations with Heading Date

**DOI:** 10.3390/plants11223116

**Published:** 2022-11-15

**Authors:** Fengyun Ma, Gina Brown-Guedira, Moonseok Kang, Byung-Kee Baik

**Affiliations:** 1Soft Wheat Quality Laboratory, United States Department of Agriculture (USDA), Agricultural Research Service (ARS)-CSWQRU, 1680 Madison Avenue, Wooster, OH 44691, USA; 2Department of Horticulture and Crop Science, The Ohio State University, 1680 Madison Avenue, Wooster, OH 44691, USA; 3Eastern Regional Small Grains Genotyping Laboratory, United States Department of Agriculture (USDA), Agricultural Research Service (ARS), Raleigh, NC 27695, USA; 4Rural Development Administration, National Institute of Crop Science, Suwon 16429, Gyeonggi, Republic of Korea

**Keywords:** wheat, allelic variation, phenology genes, genetic diversity, heading date, kompetitive allele specific PCR (KASP)

## Abstract

Wheat heading time is genetically controlled by phenology genes including vernalization (*Vrn*), photoperiod (*Ppd*) and earliness *per se* (*Eps*) genes. Characterization of the existing genetic variation in the phenology genes of wheat would provide breeding programs with valuable genetic resources necessary for the development of wheat varieties well-adapted to the local environment and early-maturing traits suitable for double-cropping system. One hundred forty-nine eastern U.S. soft winter (ESW) and 32 Korean winter (KW) wheat genotypes were characterized using molecular markers for *Vrn*, *Ppd*, *Eps* and reduced-height (*Rht*) genes, and phenotyped for heading date (HD) in the eastern U.S. region. The *Ppd-D1* and *Rht-D1* genes exhibited the highest genetic diversity in ESW and KW wheat, respectively. The genetic variations for HD of ESW wheat were largely contributed by *Ppd-B1*, *Ppd-D1* and *Vrn-D3* genes. The *Rht-D1* gene largely contributed to the genetic variation for HD of KW wheat. KW wheat headed on average 14 days earlier than ESW wheat in each crop year, largely due to the presence of the one-copy *vrn-A1* allele in the former. The development of early-maturing ESW wheat varieties could be achieved by selecting for the one-copy *vrn-A1* and *vrn-D3a* alleles in combination with *Ppd-B1a* and *Ppd-D1a* photoperiod insensitive alleles.

## 1. Introduction

Double cropping increases total crop production without the expansion of cultivation acreage. Growing soybean immediately after the harvest of winter wheat is the most common double-cropping system practiced in the United States. However, the double-crop soybean yields tend to be lower by 10–40% compared to their full-season counterparts [1]. Late planting is the major reason for the yield reduction in double-crop soybean and each day of delay in planting after mid-June reduces soybean yield by 0.65% and 1.7% in North Carolina and Maryland, respectively [2], and by 34 kg ha^−1^, 36 kg ha^−1^ and 67 kg ha^−1^ in Virginia, Georgia and Ohio, respectively [3,4,5]. The development of early-maturing wheat varieties with high yield potential is, therefore, crucial for avoiding delayed soybean planting and yield loss and for improving the profitability of the double-cropping system. The identification and acquisition of the genetic resources for the early-maturing trait, and selection of the effective molecular markers for marker-assisted selection, are prerequisites for the development of early-maturing wheat varieties. Early maturity has been a major trait considered by Korean wheat breeding programs for the development of winter wheat varieties suitable for implementing the wheat-rice double-cropping system [6]. Cho et al. [6] observed that all Korean winter (KW) wheat varieties released after 1991 carried the photoperiod insensitive allele at the photoperiod (*Ppd*)-D1 gene, which was associated with early maturity [7], and they headed on average 4 days earlier than those released before 1991. However, only 12.5–50.0% of U.S. winter wheat genotypes that carried the photoperiod insensitive allele at the *Ppd-D1* gene were observed in several worldwide and regional collections of wheat [7,8,9]. KW wheat varieties would provide bountiful genetic resources needed for the development of early-maturing U.S. winter wheat varieties.

Wheat heading time is genetically controlled by the phenology genes including vernalization (*Vrn*), *Ppd* and earliness *per se* (*Eps*) genes [8]. Vernalization requirements of wheat are controlled by three major genes including *Vrn-1*, *Vrn-2* and *Vrn-3*. The *Vrn-1* and *Vrn-3* genes are dominant in spring wheat, whereas the *Vrn-2* genes are dominant in winter wheat [10]. Among the *Vrn* genes, only *Vrn-B1* and *Vrn-D3* genes are reported to significantly influence heading date of soft winter wheat under eastern U.S. environments [9,11]. In addition, genotypes with an increased copy number of the winter allele *vrn-A1* at the *Vrn-A1* gene require a longer vernalization, thereby delaying wheat heading [12]. However, the variation in copy number of the winter allele *vrn-A1* and its effect on heading date of eastern U.S. soft winter wheat are still unknown. Wheat photoperiod response is mainly controlled by the *Ppd-A1*, *Ppd-B1* and *Ppd-D1* genes [8,12]. In general, genotypes with the photoperiod insensitive allele tend to head earlier than those with the photoperiod sensitive allele [12,13,14]. *Eps* genes influence wheat heading and flowering time when both vernalization and photoperiod requirements are satisfied and act in fine tuning the heading and flowering time [15]. However, the effects of *Eps* genes on wheat heading are usually inconsistent across environments [16]. Wheat heading time could be affected also by the reduced-height (*Rht*) genes through the gibberellin pathway [17]. Compared with the wild type, the semi-dwarf alleles *Rht-B1b* and *Rht-D1b* each have a single nucleotide substitution that causes a premature stop codon and reduces the plant’s ability to respond to gibberellic acid. Gibberellin insensitivity has been found to be associated with early heading in wheat [17].

The diversity of phenology genes has been observed in several worldwide and regional collections of wheat [7,8,17]. Substantial variation in the presence of phenology genes and their geographic distribution have been observed among continents and countries [7,8,17]. However, information is lacking on the diversity of phenology genes of eastern U.S. soft winter (ESW) wheat and their effects on wheat heading date under eastern U.S. environments. A better understanding of the allelic variation in the phenology genes of ESW wheat and their effects on heading dates would provide a valuable tool for wheat breeders in the development of early-maturing ESW wheat varieties suitable for the double-cropping system. The objectives of this study were to characterize the allelic variation in the phenology genes of ESW and KW wheat and to assess their effects on heading date under eastern U.S. environments.

## 2. Results

### 2.1. Variation in Heading Date among ESW and KW Wheat

Figure 1 shows the mean and range of HDs of 121 ESW wheat genotypes in 2015–2018, and 32 KW wheat varieties in 2017 and 2018, grown in Wooster, Ohio. The HDs of each genotype in each crop year are provided in Appendix A. Significant differences were observed in the HDs of ESW and KW wheat genotypes among the tested crop years. The HDs of ESW wheat genotypes ranged from 138 to 148, 138–143, 127–140 and 139–149 in 2015, 2016, 2017 and 2018, respectively. The average HD of ESW wheat genotypes was the shortest in 2017 (134) followed by 2016 (140), 2015 (143) and 2018 (145). Early spring temperatures in Wooster, Ohio, were warmer in 2017 than in the other years when the growing degree days after January 1 accumulated more slowly (Appendix A). HDs of KW wheat varieties ranged from 113 to 119 and 130–136 in 2017 and 2018, respectively. KW wheat varieties headed on average 14 days earlier than did ESW wheat genotypes in each crop year.

The two-way ANOVA showed that genotype, crop year and the interaction of the genotype and crop year all significantly contributed to the variation in HD (*p* < 0.0001, Appendix A). Crop year exhibited the biggest influence on HD, followed by genotype and the interaction between genotype and crop year. The HD was proved to be highly heritable as indicated by the significant correlations (r ranged from 0.52 to 0.73, *p* < 0.001, Appendix A) between HDs of ESW wheat genotypes grown in different crop years, and its high estimated heritability of 0.86. Wheat genotypes exhibited similar responses to the different environments with regards to HD.

### 2.2. Allelic Frequencies at Phenology Genes

Considerable genetic variation in the phenology gene profile was observed among ESW wheat genotypes. The phenology genes of each genotype are summarized in Appendix A. Among the 149 ESW wheat genotypes examined, 111 genotypes possessed homogeneous alleles at all the tested genes, while 38 genotypes carried heterogeneous alleles at one of the tested genes. Among the 32 KW wheat varieties examined, 30 varieties carried homogeneous alleles at all the phenology genes evaluated. The occurrence frequencies of the alleles at the phenology genes in the 149 ESW and 32 KW wheat genotypes are summarized in Table 1.

#### 2.2.1. Vernalization Genes

All ESW and KW wheat genotypes carried the winter allele at the evaluated *Vrn* genes (Table 1), which was expected since only winter wheat genotypes were included in this study. Little genetic variation was observed in the evaluated *Vrn* genes among ESW wheat (Table 1). The copy number variation (CNV) of the *vrn-A1* winter allele was significantly different between ESW and KW wheat genotypes. Most ESW wheat genotypes (94.0%) carried three or more copies of the *vrn-A1* winter allele, while most KW wheat varieties (81.3%) carried a single copy of the *vrn-A1* winter allele. Only three ESW wheat genotypes (Jamestown, Ernie and VA08MAS-369) possessed two copies or a single copy of the *vrn-A1* allele (Appendix A).

Little variation was also observed in the *Vrn-B1* gene in ESW and KW wheat. Two winter alleles (*vrn-B1-AGS2000* and *vrn-B1-Neuse*) were identified at the *Vrn-B1* gene in ESW wheat (Table 1). Most of the ESW wheat genotypes (96%) and all the KW wheat varieties (100%) carried the *vrn-B1-Neuse* allele. Only one ESW wheat variety ‘AGS 2060′ carried the *vrn-B1-AGS2000* allele. In addition, two ESW wheat genotypes were found to have a null allele at the *vrn-B1* gene.

Two winter alleles were identified at the *Vrn-D3* gene in both ESW and KW wheat genotypes (Table 1). The *vrn-D3a* allele was carried by 10.1% of ESW wheat genotypes and 46.9% of KW wheat varieties. Most ESW wheat genotypes with the *vrn-D3a* allele possessed the photoperiod sensitive allele at the *Ppd-D1* gene, while KW wheat varieties with the *vrn-D3a* allele had the *Ppd-D1a* photoperiod insensitive allele, indicating that different allelic combinations exist in ESW and KW wheat genotypes.

#### 2.2.2. Photoperiod Genes

Allelic variations in the *Ppd-A1*, *Ppd-B1* and *Ppd-D1* genes in ESW wheat genotypes were greater than those in KW wheat varieties. Most ESW wheat (56.4%) and all KW wheat (100%) carried the *Ppd-A1b* photoperiod sensitive allele. The *Ppd-B1b* photoperiod sensitive allele was carried by 73.8% of ESW wheat genotypes and 87.5% of KW wheat varieties, and was most frequent in both ESW and KW wheat genotypes. In addition, eight ESW wheat genotypes and one KW wheat variety possessed a null allele at the *Ppd-B1* gene. The *Ppd-D1a* photoperiod insensitive allele was the predominant allele at the *Ppd-D1* gene with a frequency of 59.7% among the 149 ESW wheat genotypes, followed by the *Ppd-D1b* photoperiod sensitive allele with a frequency of 16.1% (Table 1). The *Ppd-D1b-Norstar* photoperiod sensitive allele was observed in 12.8% of the 149 ESW wheat genotypes. For the KW wheat varieties, 96.9% carried the *Ppd-D1a* photoperiod insensitive allele; one KW wheat variety carried heterogeneous alleles at the *Ppd-D1* gene.

#### 2.2.3. Earliness *Per Se* Genes

The predominant alleles at the *Eps-B1* and *Eps-D1* genes were the late heading-related alleles *TaELF3-B1a* and *TaELF3-D1a* with frequencies of 92.6% and 73.2% (Table 1), respectively, among the 149 ESW wheat genotypes. All KW wheat varieties carried the late heading-related alleles *TaELF3-B1a* and *TaELF3-D1a* at both the *Eps-B1* and *Eps-D1* genes.

#### 2.2.4. Reduced Height Genes

Most ESW (63.8%) and KW (65.6%) wheat genotypes carried the wild type (tall) *Rht-B1a* allele at the *Rht-B1* gene (Table 1). The semi-dwarfing *Rht-D1b* allele was the predominant allele at the *Rht-D1* gene in ESW wheat genotypes with a frequency of 56.4%, while most KW wheat varieties (53.1%) carried the *Rht-D1a* allele at the *Rht-D1* gene. Eight ESW wheat genotypes (5.4%) and seven KW wheat varieties (21.9%) carried the tall alleles *Rht-B1a* and *Rht-D1a* at the *Rht-B1* and *Rht-D1* genes (Appendix A).

### 2.3. Genetic Diversity of Phenology Genes

The genetic diversity of *Vrn*, *Ppd*, *Eps* and *Rht* genes in the 111 ESW and 30 KW wheat genotypes, which possessed homogeneous alleles at all the tested genes, is summarized in Table 2. ESW and KW wheat had the lowest genetic diversities of the *Vrn-A1*, *Vrn-B1* and *Eps* genes but exhibited different genetic diversities of the *Vrn-D3* and *Ppd* genes (Table 2). KW wheat had the highest genetic diversity of *Vrn-D3* gene, but ESW wheat had the relatively low genetic diversity of *Vrn-D3* gene. The indices of genetic diversity at the *Ppd-A1, Ppd-B1* and *Ppd-D1* loci ranged from 0.40–0.49 and 0.00–0.18 in ESW and KW wheat, respectively (Table 2). Similar genetic diversities of the *Rht-B1* and *Rht-D1* genes were observed in both ESW and KW wheat.

### 2.4. Combined Analyses of Phenology Genes on the Heading Date of ESW Wheat over Four Crop Years and KW Wheat over Two Crop Years

Among 111 ESW wheat genotypes with homozygous alleles for each gene, 86 genotypes were evaluated for HD in four crop years and were used for testing the significance of crop year, and the *Vrn*, *Ppd*, *Eps* and *Rht* genes, and their interactions, on the HD of ESW wheat genotypes. The GLM with crop year and the phenology genes as the independent variables and HD as a dependent variable explained 92.7% of the variation in the HD of ESW wheat, with most of the variation (87.0%) explained by crop year, and a smaller amount (5.7%) explained by genetic variation (Table 3). A similarly larger influence of crop year than genetic variation on HD was also observed by Grogan et al. [8] in hard winter wheat genotypes adapted to the great plains region of the United States.

Of the genetic variation, 27.8% was contributed by the *Ppd-D1* gene, 8.9% by the *Ppd-B1* gene, 7.4% by the *Vrn-D3* gene, and 11.6% by the two-way interaction between the *Ppd-B1* gene and the *Ppd-D1* gene (Table 3). A smaller influence of genetic variation on HD was explained by the *Ppd-A1* gene (2.8%) and the *Eps-D1* gene (3.7%) and by the two-way interactions between the *Ppd-B1* (5.9%) gene or *Ppd-D1* (10.4%) gene and the *Ppd-A1* gene. The *Vrn-A1*, *Vrn-B1* and *Eps-B1* genes exhibited insignificant effects on the HD of ESW wheat genotypes, as there is limited allelic diversity at these genes (Table 2).

The influences of crop year and the phenology genes on the HD variation in KW wheat were determined using 30 KW wheat varieties (Appendix A) with homozygous alleles at each gene. Most of the HD variation of KW wheat varieties (94.7%) was explained by crop year, and a smaller amount (1.7%) was explained by the phenology genes. Of the genetic variation in the HD of KW wheat varieties, 63.4% was accounted for by the *Rht-D1* gene (Appendix A). The *Vrn*, *Ppd* and *Eps* genes did not significantly affect KW wheat HD; this mainly resulted from the lack of genetic diversity of these genes (Table 2).

### 2.5. Individual Analysis of Phenology Genes on Heading Date of ESW and KW Wheat for Each Crop Year

The GLMs with the phenology genes as the independent variables and HD as a dependent variable were tested for each crop year separately for ESW wheat genotypes (Appendix A) and KW wheat varieties (Appendix A). The best-fit GLMs varied among crop years and not all phenology genes were significant. The model explained, on average, 42.0% of the variation in the HD of ESW wheat genotypes, which ranged from a minimum of 38.0% in 2018 to a maximum of 47.1% in 2016 (Appendix A). The *Ppd-D1* gene significantly affected the HD of ESW wheat in all four crop years and accounted for, on average, 13.3% of the variation in HD, which ranged from 9.2% in 2015 to 23.5% in 2017 (Appendix A). The largest effect of the *Ppd-D1* gene on HD was observed in 2017 (Appendix A), probably due to the warm spring weather that year. The *Vrn-D3* gene exhibited a significant influence on the HD of ESW wheat only in 2015 and 2016. The interaction between the *Ppd-D1* and the *Ppd-B1* genes, and between the *Ppd-D1* and the *Vrn-D3* genes, also contributed to small variations in the HD of ESW wheat genotypes in 2018 and 2016, respectively. No significant effects of the *Vrn-A1*, *Vrn-B1*, *Ppd-A1*, *Eps-B1*, *Eps-D1*, *Rht-B1* or *Rht-D1* genes were observed on the HD of ESW wheat genotypes in any individual crop year.

Among the tested genes, only the *Rht-D1* gene showed a significant influence on the HD of KW wheat varieties in both crop years, and accounted for 21.0% of the variation in the HD in 2017 and 18.2% in 2018 (Appendix A). The other genes showed no significant influence on the HD of KW wheats in a crop year.

### 2.6. Influence of Alleles at the Vrn-D3, Ppd-B1 and Ppd-D1 Genes on the Heading Date of ESW and KW Wheat

Five genes, including the *Vrn-D3*, *Ppd-A1*, *Ppd-B1*, *Ppd-D1* and *Eps-D1* genes, were identified to significantly influence the HDs of ESW wheat genotypes in the combined analyses over four crop years (Table 3). The influences of the alleles at these five significant genes on the HD of ESW wheat were further evaluated using 86 ESW wheat genotypes, which possessed homozygous alleles at the phenology genes and were evaluated for HD in four crop years. No significant differences were observed in HD between ESW wheat genotypes carrying different alleles at the *Vrn-D3*, *Ppd-A1*, *Ppd-B1* and *Eps-D1* genes in any individual crop year. The effects of alleles at the *Ppd-D1* gene were inconsistent in four crop years (Figure 2). No significant differences in HD (except for 2017) were observed between ESW wheat genotypes carrying different alleles at the *Ppd-D1* gene. ESW wheat carrying the *Ppd-D1a* insensitive allele headed on average 2 days earlier than those carrying the *Ppd-D1b* or *Ppd-D1b-Norstar* sensitive allele in 2017 (Figure 2).

For KW wheat varieties, only the *Rht-D1* gene showed a significant influence on HD in each crop year (Appendix A). KW wheat varieties carrying the *Rht-D1a* allele headed on average 2 days earlier than those carrying the *Rht-D1b* allele in each crop year (Appendix A).

### 2.7. Influence of Allelic Combinations on the Heading Date of ESW and KW Wheat

Considering the significant influence of allelic variation in the *Vrn-D3*, *Ppd-B1* and *Ppd-D1* genes on the HD of ESW wheats (*p* < 0.001, Table 3), as well as significant differences in the allelic frequencies at the *Vrn-A1* gene between ESW and KW wheat genotypes, the influences of allelic combinations at the *Ppd-B1*, *Ppd-D1*, *Vrn-A1* and *Vrn-D3* genes on the HD of winter wheat were determined using 86 ESW wheat genotypes and 30 KW wheat varieties with homozygous alleles at these four genes. A total of 14 phenology gene profiles were observed in 86 ESW wheat genotypes (Table 4). More than half (57.0%) of 86 ESW wheat genotypes, however, possessed the same profile of *Ppd-B1b/Ppd-D1a/vrn-A1 (CNV > 2)/vrn-D3b*. Around 26.0% of ESW wheat genotypes possessed one of these four profiles: *Ppd-B1b/Ppd-D1b/vrn-A1 (CNV > 2)/vrn-D3b* (9.3%); *Ppd-B1b/Ppd-D1b-Norstart/vrn-A1 (CNV > 2)/vrn-D3b* (7.0%); *null/Ppd-D1a/vrn-A1 (CNV > 2)/vrn-D3b* (4.7%); and *Ppd-B1a-Sonora 64/Ppd-D1a/vrn-A1 (CNV > 2)/vrn-D3b* (4.7%). Nine other phenology gene profiles were observed in three or fewer genotypes. The 30 KW wheat varieties carried six phenology gene profiles at the *Ppd-B1*, *Ppd-D1*, *Vrn-A1* and *Vrn-D3* genes (Table 4). Most KW wheat varieties possessed one of these two profiles: *Ppd-B1b/Ppd-D1a/vrn-A1 (CNV = 1)/vrn-D3a* (40.0%) and *Ppd-B1b/Ppd-D1a/vrn-A1 (CNV = 1)/vrn-D3b* (36.7%). Four other phenology gene profiles were observed in three or fewer varieties.

Within ESW wheat genotypes, the average HD of 14 phenology gene profiles ranged from 140 to 144 in 2015, 139–141 in 2016, 130–136 in 2017 and 142–147 in 2018 (Table 4). Within KW wheat varieties, the average HD of six phenology gene profiles ranged from 116 to 119 in 2017 and 132–135 in 2018 (Table 4). Within ESW wheat genotypes or KW wheat varieties, relatively small variations in HD were observed among the 14 or six phenology gene profiles in each crop year, respectively, whereas each phenology gene profile exhibited considerable variation in HD among the crop years (Table 4).

Five predominant profiles of ESW wheat genotypes had the same winter alleles at the *Vrn-A1* and *Vrn-D3* genes. Among these five predominant profiles, ESW wheat genotypes carrying the profile of *Ppd-B1a-Sonora 64/Ppd-D1a/vrn-A1 (CNV > 2)/vrn-D3b* headed 1–6 days earlier than those carrying one of the remaining four profiles. Two predominant profiles of KW wheat varieties had the same alleles at the *Ppd-B1*, *Ppd-D1* and *Vrn-A1* genes. No significant differences were observed in the average HD of KW wheat varieties between different phenology gene profiles.

## 3. Discussion

### 3.1. Heading Date of ESW and KW Wheat

ESW wheat genotypes had the shortest average HDs in 2017 followed by 2016, 2015 and 2018, largely due to the warmer spring temperatures in 2017 than in the other years (Appendix A). Hu et al. [18] also reported that the HD of winter wheat was governed primarily by temperature. KW wheat varieties headed on average 14 days earlier than did ESW wheat genotypes in each crop year, indicating that genetics, in addition to crop year, have a significant influence on winter wheat HD. Würschum et al. [19] also reported that wheat HD was mainly controlled by both genotype and environment, and much less by the interaction between genotype and environment from a study of 1110 winter wheat cultivars of worldwide origin. KW wheat varieties seem to carry the genetic resources needed for the development of early-maturing ESW wheat varieties.

### 3.2. Allelic Variations in the Phenology Genes of ESW and KW Wheat

Both ESW and KW wheat showed relatively low genetic diversity in the *Vrn-A1* and *Vrn-B1* genes (Table 1 and Table 2). The small variations in the *Vrn* genes were also observed by Tessmann et al. [9] in 256 ESW wheat genotypes and Cho et al. [6] in 410 Korean wheat varieties. Increased copy number of the *Vrn-A1* gene is known to be associated with greater vernalization requirements, resulting in late heading [12]. Most ESW wheat genotypes (94.0%) carried three or more copies of the *vrn-A1* winter allele (Table 1). A single copy of the *vrn-A1* winter allele was observed in most KW wheat varieties (81.3%); however, only one (VA08MAS-369) out of 149 ESW wheat genotypes carried a single copy of the *vrn-A1* allele (Appendix A). The HDs of VA08MAS-369 in 2017 and 2018 were 135 and 146, respectively, and were longer than those of KW wheat in each crop year, largely due to the presence of the *Ppd-D1b* photoperiod sensitive allele in the former. The high frequency of a single copy of the *vrn-A1* allele in KW wheat may be related to the earlier heading of KW wheat than ESW wheat. The *vrn-B1-AGS2000* allele is known to have stronger associations with low vernalization requirements and early heading than the *vrn-B1-Neuse* allele [20]. The *vrn-B1-AGS2000* allele was observed in only one ESW wheat genotype and in none of the KW wheat varieties. The *vrn-B1-Neuse* allele was the predominant allele and was observed in 96% of ESW wheat genotypes and in 100% of KW wheat varieties. The similar frequency of the *vrn-B1-Neuse* allele in ESW and KW wheat genotypes suggests that the *Vrn-B1* gene may not be responsible for the difference in HD between ESW and KW wheat genotypes. In addition, the null allele at the *Vrn-B1* gene was detected in two ESW wheat genotypes. Tessmann et al. [9] also observed that two ESW wheat genotypes carried the null allele at the *Vrn-B1* genes in an ESW wheat collection. No significantly difference in influence on HD was observed between the null allele and other alleles at the *Vrn-B1* gene. The *vrn-D3a* allele, which is known to be associated with early heading [21], was carried by 10.1% of ESW wheat genotypes and 46.9% of KW wheat varieties. Tessmann et al. [9] observed that around 30.0% of 256 ESW wheat genotypes carried the *vrn-D3a* allele, while the frequency of the *vrn-D3a* allele varied from 3.0% to 59.5% in ESW wheat genotypes from different states, which was presented in the Appendix A reported by Tessmann et al. [9].

Allelic variations in the *Ppd* genes in ESW wheat genotypes were greater than those in KW wheat varieties. The *Ppd-A1a.1* photoperiod insensitive allele was observed in 32.9% of 149 ESW wheat genotypes, which agrees with the previous reports [9,20]. Guedira et al. [20] found that the photoperiod insensitive allele *Ppd-A1a.1* was common in ESW wheat varieties. Tessmann et al. [9] also observed that 57.4% of ESW wheat genotypes carried the *Ppd-A1a.1* allele. In contrast, the *Ppd-A1b* photoperiod sensitive allele was detected in all KW wheat varieties, which agrees with the report by Cho et al. [6], in which all 410 KW wheat genotypes carried the *Ppd-A1b* photoperiod sensitive allele. The *Ppd-A1* gene may not be responsible for the differences in HD between ESW and KW wheat genotypes as the *Ppd-A1b* photoperiod sensitive allele, which is known to be associated with late heading [14], was observed to have a high frequency in KW wheat (Table 1). The *Ppd-B1b* photoperiod sensitive allele was the predominant one observed in both ESW wheat genotypes and KW wheat varieties, and also was reported to be present in a high percentage of ESW wheat genotypes [9] and Korean wheat varieties [6]. The null allele at the *Ppd-B1* gene was detected in eight ESW wheat genotypes and one KW wheat variety. Tessmann et al. [9] observed that 7.4% of ESW wheat genotypes carried the null allele at the *Ppd-B1* gene. The association of the null allele at the *Ppd-B1* gene with HD was not evident in ESW wheat.

Grogan et al. [8] found a higher frequency of the *Ppd-A1b* photoperiod sensitive allele (98.0%) and lower frequencies of the *Ppd-B1b* photoperiod sensitive allele (57%) and the *Ppd-D1a* photoperiod insensitive allele (29.0%) in 299 U.S. great plains hard winter wheat genotypes than in ESW wheat genotypes. Guo et al. [7] observed that the predominant allele at the *Ppd-D1* gene was the photoperiod sensitive allele (*Ppd-D1b*, haplotype IV) among wheat varieties from the United States and Canada. The *Ppd-D1a* photoperiod insensitive allele was, on the other hand, predominantly observed in ESW wheat genotypes (Table 1). Tessmann et al. [9] also reported that 50.0% of ESW wheat genotypes carried the *Ppd-D1a* allele. It appears that the occurrence of the *Ppd-D1a* photoperiod insensitive allele varies among different wheat classes in the United States. The *Ppd-D1a* photoperiod insensitive allele was predominantly observed in KW wheat with a frequency of 96.9%, which agrees with the report by Guo et al. [7] who found that haplotype I (*Ppd-D1a* photoperiod insensitive allele) was most common in Asian wheat varieties. Cho et al. [6] also observed that all KW wheat varieties developed since 1991 carried the *Ppd-D1a* photoperiod insensitive allele.

The *TaELF3-B1a* and *TaELF3-D1a* alleles, known to be associated with late heading [15], were predominantly observed in both ESW and KW wheat genotypes, suggesting that the *Eps-B1* and *Eps-D1* genes may not be responsible for the difference in HD between ESW and KW wheat genotypes.

The semi-dwarf alleles *Rht-B1b* and *Rht-D1b* were present in ESW and KW wheat at similar frequencies to those observed by Guedira et al. [22] in 247 ESW wheat varieties released before 2008 and Cho et al. [6] in 410 KW wheat genotypes. Wilhelm et al. [17] and Grogan et al. [8] reported that the *Rht-B1b* and *Rht-D1b* alleles could promote wheat heading probably through their significant associations with the reduction in gibberellin sensitivity. The similar frequencies of the *Rht-B1b* and *Rht-D1b* alleles in ESW and KW wheat genotypes suggest that the *Rht-B1* and *Rht-D1* genes may not contribute to the difference in HD between ESW and KW wheat genotypes.

### 3.3. Influences of Phenology Genes on the Heading Date of ESW and KW Wheat

Five genes, including *Vrn-D3*, *Ppd-A1*, *Ppd-B1*, *Ppd-D1* and *Eps-D1*, were identified to significantly influence the HD of ESW wheat (Table 3). These results agree with the report by Huang et al. [11], who found that the *Ppd-A1*, *Ppd-B1*, *Ppd-D1* and *Vrn-D3* genes significantly affected the HD of ESW wheat under eastern U.S. environments. The low levels of genetic diversity in the *Vrn-A1*, *Vrn-B1* and *Eps-B1* genes explains their insignificant influences on ESW wheat HD. Thus, additional variation in the HD of ESW wheat genotypes could be achieved by increasing the genetic diversities in the *Vrn* and *Eps* genes and the introduction of the early heading-related alleles of those genes, such as the one-copy *vrn-A1* winter allele and the *vrn-B1-AGS2000* allele.

No significant differences were observed in HD between ESW wheat genotypes carrying different alleles at the *Ppd-A1*, *Ppd-B1* and *Eps-D1* genes in any individual crop year. These results agree with the report by Tessmann et al. [9], in which ESW wheat genotypes carrying the different alleles at the *Ppd-B1* gene exhibited no significant difference in HD under eastern U.S. environments. The low frequency of the photoperiod insensitive allele at the *Ppd-B1* gene may contribute to its insignificant influence on HD. Griffiths et al. [16] observed that the effects of *Eps* genes on wheat heading were inconsistent across environments. Additionally, no significant differences in HD were also observed between ESW wheat genotypes carrying the *vrn-D3a* allele or the *vrn-D3b* allele at the *Vrn-D3* gene. The allelic variation in the *Ppd-D1* gene showed variable influence on HD among crop years, suggesting differential expression under varying climatic conditions, especially temperature. The early heading of ESW wheat genotypes with the photoperiod insensitive allele *Ppd-D1a* agree with the reports by Grogan et al. [8] and Whittal et al. [23]. These results indicate that the effects of alleles at phenology genes on ESW wheat HD are inconsistent across crop years and the existing genetic resources of ESW wheat may not be sufficient to provide the early maturing trait in the development of early-maturing ESW wheat varieties.

The *Vrn*, *Ppd* and *Eps* genes were found to have insignificant effects on KW wheat HD, considering their lack of genetic variations. The *Rht-D1* gene had the highest genetic variation in KW wheat and its variation might explain most of the genetic variation in KW wheat HD (Appendix A). Grogan et al. [8] found that the *Rht-B1* gene significantly affected the HD of hard winter wheat genotypes grown in the U.S. great plains, but not the *Rht-D1* gene. In this study, the *Rht-B1* and *Rht-D1* genes had insignificant influences on the HD of ESW wheat genotypes under the tested environments. KW wheat varieties carrying the *Rht-D1a* allele headed on average 2 days earlier than those carrying the *Rht-D1b* allele in each crop year (Appendix A). Tessmann et al. [9] also observed a slight decrease (0.6%) in the HD of ESW wheat genotypes with the *Rht-D1a* allele compared to those with the *Rht-D1b* allele under eastern U.S. environments. Several other studies reported that the presence of the *Rht-B1b* and *Rht-D1b* alleles in wheat could lead to early heading through their significant associations with the reduction in gibberellin sensitivity [8,17]. It is not evident why the *Rht-D1a* allele shows an association with early heading in winter wheat.

### 3.4. Influence of Allelic Combinations on the Heading Date of ESW and KW Wheat

A total of 14 and 6 phenology gene profiles at four major genes (*Ppd-B1*, *Ppd-D1*, *Vrn-A1* and *Vrn-D3*) were observed in ESW and KW wheat, respectively. A phenology gene profile (*Ppd-B1b/Ppd-D1a/vrn-A1* (three or more copies)/*vrn-D3b*) was predominantly observed in 57.0% of ESW genotypes, while 76.7% of KW varieties carried one of these two phenology gene profiles: *Ppd-B1b/Ppd-D1a/vrn-A1* (one copy)/*vrn-D3a* or *Ppd-B1b/Ppd-D1a/vrn-A1* (one copy)/*vrn-D3b*. Within each population, relatively small variations in HD were observed among the phenology gene profiles in each crop year, whereas each phenology gene profile exhibited considerable variation in HD among the crop years (Table 4). Wheat genotypes adapted to a specific location have HDs appropriate to the local climatic conditions to ensure the maximum yield potential. The copy number variation in the *Vrn-A1* gene was the major difference in the predominant phenology gene profiles between ESW and KW wheat genotypes and might be responsible for the differences in HDs between ESW and KW wheat genotypes under eastern U.S. environments. The introduction of the one-copy *vrn-A1* winter allele, and the increased frequencies of the *vrn-D3a* winter allele and the photoperiod insensitive alleles at the *Ppd-B1* and *Ppd-D1* genes should be considered in the development of early-maturing ESW wheat varieties suitable for wheat–soybean double-cropping systems.

## 4. Materials and Methods

### 4.1. Materials

Two populations, including 149 eastern U.S. soft winter (ESW) wheat genotypes and 32 Korean winter (KW) wheat varieties, were used in this study. The ESW wheat genotypes were obtained from over 20 ESW wheat breeding programs, while the seeds of KW wheat varieties were obtained from the International Maize and Wheat Improvement Center (CIMMYT). The heading dates for ESW wheat genotypes and KW wheat varieties were determined over two consecutive crop years (2017 and 2018) in Wooster, Ohio (40.8051° N, 81.9351° W). ESW wheat genotypes were additionally phenotyped for heading date in 2015 and 2016 in Wooster, Ohio (Table 5 and Appendix A). A complete list of ESW and KW wheat genotypes can be found in Appendix A. Wheat genotypes were planted in a randomized complete block design with two replications. Heading date (HD) was recorded as the Julian date when the 50% of the spikes in a plot had fully emerged from the flag leaf sheath.

### 4.2. Genetic Evaluation

Genomic DNA was extracted from 10–15-day-old hypocotyls of germinating seeds using DNAzol ES (Extra Strength) reagent (Molecular Research Center, Inc., Cincinnati, OH, USA) according to the manufacturer’s protocol. Molecular marker assays for phenology genes were conducted at the USDA-ARS Eastern Regional Small Grain Genotyping Laboratory, Raleigh, NC, USA. Kompetitive Allele Specific PCR (KASP) markers developed from published sequences were used to distinguish alleles at the *Vrn-A1*, *Vrn-B1*, *Ppd-A1*, *Ppd-B1*, *Ppd-D1*, *Eps-B1*, *Eps-D1*, *Rht-B1* and *Rht-D1* genes (Table 6 and Appendix A). The recessive allele *vrn-A1* was distinguished from the dominant alleles *Vrn-A1a* and *Vrn-A1b* using molecular markers wMAS000033 and wMAS000035 [24], respectively. Two molecular markers (vrn-A1exon4_C/T and vrn-A1exon7_G/A) were used to determine copy number variation (CNV) of winter allele *vrn-A1* [12]. Three markers (TaVrn-B1_D-I, wMAS000037, and Vrn-B1_C) were used to distinguish the dominant alleles *Vrn-B1a*, *Vrn-B1b* [25], and *Vrn-B1c* [26] at the *Vrn-B1* gene. Molecular marker (TaVrn-B1_1752) was used to detect an A/G polymorphism in intron 1 of the recessive allele *vrn-B1* [20] and to distinguish alleles (*vrn-B1-Neuse* and *vrn-B1-AGS2000*). Genotypes were considered to carry a null allele at the *Vrn-B1* gene if no PCR amplification was detected using the molecular marker TaVrn-B1_1752. The primer pair VRN-D3-F6/VRN-D3-R8 was used to distinguish alleles at the *Vrn-D3* gene according to the method described by Chen et al. [21].

Photoperiod insensitive allele *Ppd-A1a.1* was distinguished with the marker TaPpd-A1prodel [14], which detects a deletion characteristic of the insensitive allele. Alleles at the *Ppd-B1* gene were distinguished using two markers: wMAS000027, which detects the ‘Chinese Spring’-type insensitive allele with a truncated copy and TaPpdBJ003, which identifies the ‘Sonora 64′-type insensitive allele with an inter copy junction [12]. Wheat genotypes were considered to carry the null allele at the *Ppd-B1* gene if no amplification occurred with the markers. Photoperiod sensitive and insensitive alleles at the *Ppd-D1* gene were distinguished using two markers: wMAS000024, which detects a deletion upstream of the coding region responsible for the photoperiod insensitive phenotype [13], and TaPpdDD002, which identifies the ‘Norstar’-type sensitive allele [27]. Mutations at the *Eps-B1* and *Eps-D1* loci were detected using markers TaELF3-B1 Kasp and TaELF3-D1 Kasp2, respectively [15]. Mutations at the *Rht-B1* and *Rht-D1* loci were genotyped using wMAS000001 and wMAS000002 [28], respectively.

### 4.3. Genetic Diversity

The genetic diversity at each locus was calculated as described by Nei [29] using the equation: H = 1−∑ *p_i_*^2^ (where H is Nei’s genetic diversity index, and *p_i_* is the frequency of the *i*th allele at that locus). The frequencies of alleles were counted only for those genotypes with a homogeneous allele at all the tested loci and used for calculating the genetic diversity.

### 4.4. Statistical Analysis

Statistical analyses were performed using Statistical Analysis System software (SAS Institute, Cary, NC, USA). A general linear model (GLM) was used to determine the effects of crop years, and the vernalization, photoperiod, earliness *per se* and reduced-height genes, and their interactions, on the HD of ESW and KW wheat genotypes in a combined analysis over four crop years for the former and two crop years for the latter, or in each crop year for both. The sum of squares was used to estimate the proportion of total variance contributed by crop year and each gene, and the proportion of genetic variance contributed by each gene [8]. A least significant difference (LSD) test was used to determine the differences in the effects of crop year, individual allele at each gene, and phenology gene profiles on the HD of ESW and KW wheat genotypes with homozygous alleles at the phenology genes. Pearson’s linear correlation analysis was conducted to determine the relationships between HDs of ESW wheat genotypes grown in different crop years. A two-way analysis of variance (ANOVA) test was conducted to evaluate the effects of crop years and genotype and their interaction effects on HD using 121 ESW wheat genotypes. Broad sense heritability (*H^2^*) was calculated over four crop years in terms of mean squares (MS): *H^2^* = (MS_Genotype_ − MS_Genotype_ × _Year_)/MS_Genotype_ [30].

## 5. Conclusions

The genetic effects on the HD of ESW wheat genotypes were largely explained by the *Ppd-B1*, *Ppd-D1* and *Vrn-D3* genes. The *Rht-D1* gene had the highest genetic diversity in KW wheat. ESW wheat genotypes exhibited, on average, an HD 14 days later than that of KW wheat varieties in each crop year, largely due to the absence of a single copy of the *vrn-A1* allele in the former. A phenology gene profile (*Ppd-B1b/Ppd-D1a/vrn-A1 (CNV > 2)/vrn-D3b*) was predominantly observed in ESW wheat genotypes with a frequency of 57.0%. Two phenology gene profiles (*Ppd-B1b/Ppd-D1a/vrn-A1 (CNV = 1)/vrn-D3a* and *Ppd-B1b/Ppd-D1a/vrn-A1 (CNV = 1)/vrn-D3b*) were predominantly observed in KW wheat varieties, making up 76.7% of the total observed profiles. The development of early-maturing ESW wheat varieties could be achieved by selecting for the one-copy *vrn-A1* and *vrn-D3a* winter alleles in combination with photoperiod insensitive alleles at the *Ppd-B1* and *Ppd-D1* genes. These results enhance our understanding of the effects of phenology genes on the HD of winter wheat under eastern U.S. environments and could be useful for the development of early-maturing wheat varieties suitable for wheat–soybean double-cropping systems and wheat varieties well-adapted to the current and future climate conditions.

## Figures and Tables

**Figure 1 plants-11-03116-f001:**
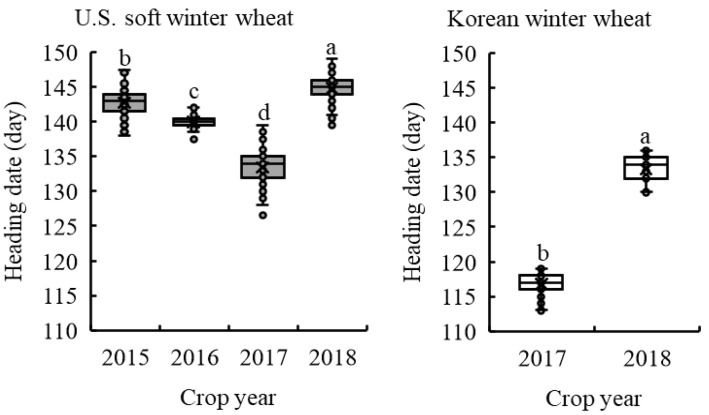
Mean heading dates of 121 eastern U.S. soft winter wheat genotypes grown in 2015–2018 and 32 Korean winter wheat varieties grown in 2017 and 2018 in Wooster, Ohio. x indicates the mean. Means with different letters are significantly different at *p* < 0.05.

**Figure 2 plants-11-03116-f002:**
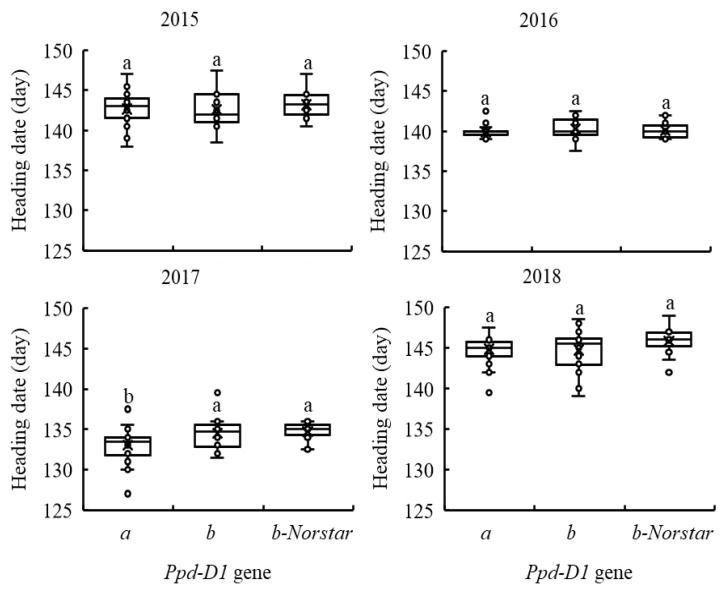
Heading dates of 86 eastern U.S. soft winter wheat genotypes with different alleles at the *Ppd-D1* gene in four crop years. *b-Norstar*, ‘Norstar’-type sensitive allele. x indicates the mean. Means with different letters are significantly different at *p* < 0.05.

**Table 1 plants-11-03116-t001:** Frequency of different alleles at the vernalization (*Vrn*), photoperiod (*Ppd*), earliness *per se* (*Eps*) and reduced-height (*Rht*) genes in 149 eastern U.S. soft winter wheat genotypes and 32 Korean winter wheat varieties.

Gene	Allele	Frequency (%)	
		U.S. Soft Winter Wheat	Korean Winter Wheat
*Vrn-A1*	*vrn-A1*, CNV ^a^ > 2	94.0	12.5
	*vrn-A1*, CNV = 2	1.3	0.0
	*vrn-A1*, CNV = 1	0.7	81.3
	heterogeneous	4.0	6.3
*Vrn-B1*	*vrn-B1-Neuse*	96.0	100.0
	*vrn-B1-AGS2000*	0.7	0.0
	null	1.3	0.0
	heterogeneous	2.0	0.0
*Vrn-D3*	*vrn-D3a*	10.1	46.9
	*vrn-D3b*	89.9	53.1
*Ppd-A1*	*Ppd-A1a.1*	32.9	0.0
	*Ppd-A1b*	56.4	100.0
	heterogeneous	10.7	0.0
*Ppd-B1*	*Ppd-B1a-Chinese Spring*	12.8	6.3
	*Ppd-B1a-Sonora 64*	6.7	3.1
	*Ppd-B1b*	73.8	87.5
	null	5.4	3.1
	heterogeneous	1.3	0.0
*Ppd-D1*	*Ppd-D1a*	59.7	96.9
	*Ppd-D1b*	16.1	0.0
	*Ppd-D1b-Norstar*	12.8	0.0
	heterogeneous	11.4	3.1
*Eps-B1*	*TaELF3-B1a*	92.6	100.0
	*TaELF3-B1b*	3.4	0.0
	heterogeneous	4.0	0.0
*Eps-D1*	*TaELF3-D1a*	73.2	100.0
	*TaELF3-D1b*	19.5	0.0
	heterogeneous	7.4	0.0
*Rht-B1*	*Rht-B1a*	63.8	65.6
	*Rht-B1b*	33.6	31.3
	heterogeneous	2.7	3.1
*Rht-D1*	*Rht-D1a*	38.9	53.1
	*Rht-D1b*	56.4	46.9
	heterogeneous	4.7	0.0

^a.^ CNV, copy number variation.

**Table 2 plants-11-03116-t002:** Genetic diversity of the vernalization (*Vrn*), photoperiod (*Ppd*), earliness *per se* (*Eps*) and reduced-height (*Rht*) genes in eastern U.S. soft winter wheat genotypes and Korean winter wheat varieties.

Gene	Nei’s Genetic Diversity	
	U.S. Soft Winter Wheat	Korean Winter Wheat
*Vrn-A1*	0.02	0.23
*Vrn-B1*	0.05	0.00
*Vrn-D3*	0.16	0.50
*Ppd-A1*	0.44	0.00
*Ppd-B1*	0.40	0.18
*Ppd-D1*	0.49	0.00
*Eps-B1*	0.09	0.00
*Eps-D1*	0.31	0.00
*Rht-B1*	0.43	0.42
*Rht-D1*	0.47	0.50

**Table 3 plants-11-03116-t003:** The influence of crop year, and the vernalization (*Vrn*), photoperiod (*Ppd*), earliness *per se* (*Eps*) and reduced-height (*Rht*) genes and their interactions, on the heading date of eastern U.S. soft winter wheat genotypes in four crop years.

Source of Variation	Degrees of Freedom	Mean Square	Proportion of Total Variance (%)	Proportion of Genetic Variance (%)
Crop year	3	2119.6 ***	87.0	
*Vrn-A1*	2	2.5	0.1	1.2
*Vrn-B1*	2	0.4	0.0	0.2
*Vrn-D3*	1	30.3 ***	0.4	7.4
*Ppd-A1*	1	11.7 *	0.2	2.8
*Ppd-B1*	3	12.2 ***	0.5	8.9
*Ppd-D1*	2	57.3 ***	1.6	27.8
*Eps-B1*	1	5.8	0.1	1.4
*Eps-D1*	1	15.3 **	0.2	3.7
*Rht-B1*	1	3.5	0.0	0.8
*Rht-D1*	1	2.7	0.0	0.6
*Ppd-A1* * *Ppd-B1*	3	8.1 **	0.3	5.9
*Ppd-A1* * *Ppd-D1*	1	42.9 ***	0.6	10.4
*Ppd-B1* * *Ppd-D1*	4	11.9 ***	0.7	11.6
*Ppd-A1* * *Ppd-B1* * *Ppd-D1*	1	0.1	0.0	0.0
Error	262	2.0	7.3	

The sum of squares was used to calculate the proportion of total variance contributed by crop years and each gene, and the proportion of genetic variance by each gene. Eighty-six eastern U.S. soft winter wheat genotypes with homozygous alleles at each gene were evaluated for heading date in four crop years and included in the analyses of variation. *** significant at *p* < 0.001, ** significant at *p* < 0.01, * significant at *p* < 0.05.

**Table 4 plants-11-03116-t004:** Phenology gene profiles of 86 eastern U.S. soft winter wheat genotypes and 30 Korean winter wheat varieties carrying homozygous alleles at the *Ppd-B1*, *Ppd-D1*, *Vrn-A1* and *Vrn-D3* genes sorted by frequency and their heading dates.

Allele				Frequency (%)	Heading Date (Day)			
*Ppd-B1*	*Ppd-D1*	*vrn-A1* CNV ^a^	*vrn-D3*		2015	2016	2017	2018
U.S. soft winter wheat								
*b*	*a*	>2	*b*	57.0	143	140	133	145
*b*	*b*	>2	*b*	9.3	144	141	135	146
*b*	*b-Nor ^d^*	>2	*b*	7.0	144	140	136	147
*null*	*a*	>2	*b*	4.7	143	140	134	146
*a-S64 ^e^*	*a*	>2	*b*	4.7	140	139	130	143
*b*	*b*	>2	*a*	3.5	141	139	134	144
*a-CS*	*a*	>2	*b*	3.5	143	141	133	145
*a-CS ^f^*	*b-Nor*	>2	*b*	2.3	142	139	134	144
*b*	*b-Nor*	>2	*a*	2.3	142	140	133	144
*a-CS*	*b*	>2	*a*	1.2	141	139	130	142
*a-CS*	*b*	>2	*b*	1.2	144	141	136	146
*null*	*b*	>2	*b*	1.2	142	140	133	142
*null*	*b-Nor*	>2	*b*	1.2	144	141	135	146
*b*	*b*	1	*a*	1.2	143	139	135	146
				LSD (0.05) ^c^	4	2	4	4
Korean winter wheat								
*b*	*a*	1	*a*	40.0	nd ^b^	nd	116	133
*b*	*a*	1	*b*	36.7	nd	nd	117	133
*b*	*a*	>2	*b*	10.0	nd	nd	118	134
*a-CS*	*a*	1	*b*	6.7	nd	nd	119	135
*b*	*a*	>2	*a*	3.3	nd	nd	117	132
*null*	*a*	1	*a*	3.3	nd	nd	119	135
				LSD (0.05)			4	4

^a.^ CNV, copy number variation. ^b.^ nd, not determined. ^c.^ LSD, least significant difference (at *p* < 0.05). ^d.^ *b-nor*, ‘Norstar’-type sensitive allele. ^e.^ *a-S64*, ‘Sonora 64′-type insensitive allele. ^f.^ *a-CS*, ‘Chinese Spring’-type insensitive allele.

**Table 5 plants-11-03116-t005:** Populations evaluated for heading date.

Population	Crop Year	Growing Location	Number of Entries
Eastern U.S. soft winter wheat ^a^	2015	Wooster, Ohio	125
	2016	Wooster, Ohio	149
	2017	Wooster, Ohio	149
	2018	Wooster, Ohio	138
Korean winter wheat	2017	Wooster, Ohio	32
	2018	Wooster, Ohio	32

^a.^ A total of 121 ESW wheat genotypes were phenotyped for heading date over four consecutive crop years.

**Table 6 plants-11-03116-t006:** Alleles of vernalization (*Vrn*), photoperiod (*Ppd*), earliness *per se* (*Eps*) and reduced-height (*Rht*) genes governing the heading date of wheat.

Gene	Allele	Marker ID	Allele Effect	Reference
*Vrn-A1*	*Vrn-A1a*	wMAS000033	spring growth habit	[24]
	*Vrn-A1b*	wMAS000035	spring growth habit	[24]
	*vrn-A1,* CNV ^a^ > 2	vrn-A1exon4	winter growth habit, late heading	[12]
	*vrn-A1,* CNV = 2	vrn-A1exon7	winter growth habit, late heading	[12]
	*vrn-A1,* CNV = 1	vrn-A1exon7	winter growth habit, early heading	[12]
*Vrn-B1*	*Vrn-B1a*	Vrn-B1_I_D	spring growth habit	[25]
	*Vrn-B1b*	wMAS000037	spring growth habit	[25]
	*Vrn-B1c*	Vrn-B1_C	spring growth habit	[26]
	*vrn-B1-Neuse*	TaVrn-B1_1752	winter growth habit, late heading, Neuse-type	[20]
	*vrn-B1-AGS2000*	TaVrn-B1_1752	winter growth habit, early heading, AGS2000-type	[20]
*Vrn-D3*	*vrn-D3a*	VRN-D3-F6/VRN-D3-R8	winter growth habit with early heading	[21]
	*vrn-D3b*	VRN-D3-F6/VRN-D3-R8	winter growth habit with late heading	[21]
*Ppd-A1*	*Ppd-A1a.1*	Ppd-A1prodel	photoperiod insensitive, early heading	[14]
	*Ppd-A1b*	Ppd-A1prodel	photoperiod sensitive, late heading	[14]
*Ppd-B1*	*Ppd-B1a-Chinese Spring*	wMAS000027	photoperiod insensitive, early heading	[12]
	*Ppd-B1a-Sonora 64*	TaPpdBJ003	photoperiod insensitive, early heading	[12]
	*Ppd-B1b*	wMAS000027	photoperiod sensitive, late heading	[12]
*Ppd-D1*	*Ppd-D1a*	wMAS000024	photoperiod insensitive, early heading	[13]
	*Ppd-D1b*	wMAS000024	photoperiod sensitive, late heading	[13]
	*Ppd-D1b-Norstar*	TaPpdDD002	photoperiod sensitive, late heading	[27]
*Eps-B1*	*TaELF3-B1a*	TaELF3-B1 Kasp	late heading and flowering	[15]
	*TaELF3-B1b*	TaELF3-B1 Kasp	early heading and flowering	[15]
*Eps-D1*	*TaELF3-D1a*	TaELF3-D1 Kasp2	late heading and flowering	[15]
	*TaELF3-D1b*	TaELF3-D1 Kasp2	early heading and flowering	[15]
*Rht-B1*	*Rht-B1a*	wMAS000001	tall	[28]
	*Rht-B1b*	wMAS000001	semi-dwarf	[28]
*Rht-D1*	*Rht-D1a*	wMAS000002	tall	[28]
	*Rht-D1b*	wMAS000002	semi-dwarf	[28]

^a.^ CNV, copy number variation.

## Data Availability

All of the data reported in this manuscript are provided in the Appendix A.

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
