# Peer review of "Allelic Variations in Phenology Genes of Eastern U.S. Soft Winter and Korean Winter Wheat and Their Associations with Heading Date"

_plants, 2022, doi:10.3390/plants11223116_

Round 1

Reviewer 1 Report

The study presented in the manuscript is both elaborated and professionally interpreted. Through the complex analysis of four key phenology genes, the authors provide a valuable marker-assisted selection technique for early maturing wheat breeding. However, the phenology data is not fully complete in the case of the Korean winter wheat genotypes (only two crop years’ data are available), the study still holds considerable novelty value.

The applied methodology is appropriate and sophisticated. The obtained results are clear and well-interpreted. Statistics are approved.

Some minor mistakes should be corrected: line 176 -'low/lowest", line 186-187- four crop years data are available only for ESW genotypes.

The manuscript is recommended for acceptance without any (or very minor) changes.

Author Response

Reviewer 1

  1. Some minor mistakes should be corrected: line 176 -'low/lowest", line 186-187- four crop years data are available only for ESW genotypes.

Response: We have revised the manuscript as suggested.

Line 179: ‘low’ was changed to 'lowest'.

Lines 189-190: ‘the heading date of ESW and KW wheat over four crop years’ was changed to ‘the heading date of ESW wheat over four crop years and KW wheat over two crop years’.

Reviewer 2 Report

The authors compared the heading dates of wheat varieties from the Eastern U.S. and Korea, and analyzed the diversity of Vrn1, Ppd1, Eps, and Rht1 genes, aiming to provide early maturing wheat varieties for the double-cropping system in the United States, but the authors need to address the following issues:

1. The days to heading cannot completely represent the days to maturity, and the author should compare the differences in the maturation date of varieties

2. There are also VRNT, VRN2, FT, PHY-C, SOC, Hd1, GI, and TOC-1D genes that affect heading date. Why the authors did not analyze them?

3. The author compared the differences in heading dates. I would like to know the differences in other traits under different gene combinations, especially the grain number per spike, one thousand grain weight, and effective tiller.

4. Has the study identified any wheat varieties in line with the American double-cropping system or created such germplasm?

Author Response

Reviewer 2

The authors compared the heading dates of wheat varieties from the Eastern U.S. and Korea, and analyzed the diversity of Vrn1, Ppd1, Eps, and Rht1 genes, aiming to provide early maturing wheat varieties for the double-cropping system in the United States, but the authors need to address the following issues:

Response: The objectives of this study as described in the manuscript were to characterize the allelic variation in the phenology genes of ESW and KW wheat and to assess their effects on heading date under eastern U.S. environments, which will lead to the identification of the genes, genetic resources and potential molecular markers useful for the development of early maturing ESW wheat suitable for double-cropping system.

Please see our response to your comments below.

  1. The days to heading cannot completely represent the days to maturity, and the author should compare the differences in the maturation date of varieties

Response: We agree that the days to heading may not fully represent the days to maturity, but a strong relationship exists between them. Whitta et al. (2018) observed that the heading date of winter wheat showed a strong relationship with the maturation date (r = 0.78, p<0.001) for 203 winter wheat varieties (https://doi.org/10.1371/journal.pone.0203068). The early-heading wheat tends to mature early.

We observed that both heading date and maturing date of Korean winter wheat varieties were around 14 days earlier than those ESW wheat genotypes in the tested crop years.

  1. There are also VRNT, VRN2, FT, PHY-C, SOC, Hd1, GI, and TOC-1D genes that affect heading date. Why the authors did not analyze them?

Response: As we mentioned in the introduction section, wheat heading time is mainly genetically controlled by the phenology genes including Vrn, Ppd, Eps and Rht genes (Grogan get al. 2016, https://doi.org/10.1371/journal.pone.0152852). In addition, only four genes including Ppd-A1, Ppd-B1, Ppd-D1 and Vrn-D3 have been reported to have significant influences on ESW wheat heading date under eastern U.S. environments as reported by Huang et al. (2018, https://doi.org/10.1007/s10681-018-2199-y) using 90K SNP makers. Therefore, this study focused on those key phenology genes influencing ESW wheat heading, which would provide meaningful information for ESW wheat breeders for screening breeding lines with early heading trait.

The VRNT, VRN2, FT, PHY-C, SOC, Hd1, GI, and TOC-1D genes were not tested due to their insignificant influences on ESW wheat heading under eastern U.S. environments as reported by Huang et al. (2018) using 90K SNP markers.

  1. The author compared the differences in heading dates. I would like to know the differences in other traits under different gene combinations, especially the grain number per spike, one thousand grain weight, and effective tiller.

Response: As this study focused on the phenology genes controlling heading date, we didn’t collect grain number per spike, thousand grain weight and effective tiller during the test crop years.

  1. Has the study identified any wheat varieties in line with the American double-cropping system or created such germplasm?

Response: No ESW wheat varieties were identified to have heading date considerably earlier than others and so be fully desirable for the double-cropping system in the region, An effort to develop the early-maturing ESW wheat germplasms has been initiated using the genetic resources and molecular markers identified from this study.

Reviewer 3 Report

Heading time is mainly controlled by vernalization (Vrn), photoperiod (Ppd) and earliness per se (Eps) genes. It is necessary to detect the genetic variation in the phenology genes for providing valuable genetic resources for the development of wheat varieties well-adapted to the local environment and early-maturing traits suitable for double-cropping system. In this study, a total of 181 wheat cultivars were genotyped using molecular markers for Vrn, Ppd, Eps and reduced-height (Rht) genes, and phenotyped for heading date (HD) in eastern U.S. region for four years. This study provided the information on selection of valuable genetic resources and reliable molecular markers, but some concerns need to be resolved:

1.     Lines 59-61: the authors should be provided the detained introduction about Vrn-A1, Vrn-B1 and Vrn-D3 in order to make readers understand why select these three loci in this study. In addition, the Vrn-D1 and Vrn-B3 are important vernalization loci, why you did not select them in this study.

2.     Line 91: supplement unit of heading date “d”, including in Figures 1 and 2.

3.     Lines 104-110: the results of ANOVA and correlation analysis should be supplemented in the Supplementary materials.

4.     Line 133: author used “locus” in tables 1 and 2, but used “gene” in the Result section, which need to be kept consistent.

5.     Lines 174, 189 and 247: the number of the homozygous genotypes is not consistent in different analysis, for example, in Line 174 the number is 111, but in Line 189 the number is 86

6.     Lines 472: the detailed primer sequence information of KASP markers should be supplemented in this study

7.     Lines 527-547, the Conclusion need to be simplified.

Author Response

Reviewer 3

  1. Lines 59-61: the authors should be provided the detained introduction about Vrn-A1, Vrn-B1 and Vrn-D3 in order to make readers understand why select these three loci in this study. In addition, the Vrn-D1 and Vrn-B3 are important vernalization loci, why you did not select them in this study.

Response: We have added the explanation on the selection of Vrn-A1, Vrn-B1 and Vrn-D3 to the introduction section. (Lines 61-66)

The Vrn-B1 and Vrn-D3 gene were included in this study because among Vrn genes, only Vrn-B3 and Vrn-D3 genes exhibited significant influence on ESW wheat heading under eastern U.S. environments as reported by Tessmann et al (2019, https://doi.org/10.2135/cropsci2018.08.0492) and Huang et al. (2018, https://doi.org/10.1007/s10681-018-2199-y), respectively.

In addition, genotypes with an increased copy number of the winter allele vrn-A1 at the Vrn-A1 gene require a longer vernalization, thereby delaying wheat heading (Díaz et al., 2012, https://doi.org/10.1371/journal.pone.0033234). However, the frequency of different copy numbers of the winter allele vrn-A1 and its effect on heading date of eastern U.S. soft winter wheat are still unknown. Therefore, this study focused on these three key Vrn genes and compared these genes between eastern U.S. wheat and early-maturing Korean winter wheat, which would provide meaningful information for ESW wheat breeders for screening breeding lines with early heading trait.

We agree that the Vrn-D1 and Vrn-B3 are also important vernalization loci, however, they were not tested in this study as their insignificant influences on ESW wheat heading as reported by Huang et al. (2018) using 90K SNP markers. In addition, the low genetic diversity of Vrn-D1 was observed in ESW wheat based on 2021 ESW wheat Nursery Marker Report, where all 603 ESW wheat lines carried winter allele vrn-D1 at Vrn-D1 gene (data available at https://www.ars.usda.gov/southeast-area/raleigh-nc/plant-science-research/docs/small-grains-genotyping-laboratory/regional-nursery-marker-reports/).

  1. Line 91: supplement unit of heading date “d”, including in Figures 1 and 2.

Response: We have added the unit of heading date ‘day’ to Figures 1 and 2.

  1. Lines 104-110: the results of ANOVA and correlation analysis should be supplemented in the Supplementary materials.

Response: We have added two Supplemental Tables S2 and S3 showing the results of ANOVA and correlation analysis, respectively. (Supplemental Tables S2 and S3)

  1. Line 133: author used “locus” in tables 1 and 2, but used “gene” in the Result section, which need to be kept consistent.

Response: We have changed ‘Locus’ to ‘gene’ in Tables 1 and 2. (Tables 1 and 2)

  1. Lines 174, 189 and 247: the number of the homozygous genotypes is not consistent in different analysis, for example, in Line 174 the number is 111, but in Line 189 the number is 86.

Response: We have revised to make it clear.

In this study, 111 out of 149 ESW wheat genotypes possessed homogeneous alleles at all the tested genes and thus only 111 ESW wheat genotypes were used for analyzing the genetic diversity (Table 2). We have described 111 ESW wheat genotypes in Lines 118-119 and 177-178.

Among these 111 ESW wheat genotypes, 86 genotypes were evaluated for heading date across all four crop years and thus only 86 ESW wheat genotypes were used for combined analysis or individual analysis of phenology genes on heading date to easily compare the results among the different analyses. Revised to make it clear. (Lines 191-193)

  1. Lines 472: the detailed primer sequence information of KASP markers should be supplemented in this study

Response: We included Supplemental Table S7 describing the detailed primer sequences of KAPS markers used in this study.

  1. Lines 527-547, the Conclusion need to be simplified.

Response: Deleted a few sentences and simplified Conclusion as suggested. (Lines 531-546)

Round 2

Reviewer 3 Report

Accept in present form.